# Elliptic Curve-Based Query Authentication Protocol for IoT Devices Aided by Blockchain

**DOI:** 10.3390/s23031371

**Published:** 2023-01-26

**Authors:** Stefania Loredana Nita, Marius Iulian Mihailescu

**Affiliations:** 1Department of Computers and Cybernetic Security, “Ferdinand I” Military Technical Academy, 050141 Bucharest, Romania; 2Scientific Research Center in Mathematics and Computer Science, “Spiru Haret” University, 030045 Bucharest, Romania

**Keywords:** authentication, IoT, elliptic curves, blockchain

## Abstract

Digital transformation has increased its proportion in the last few years and the Internet-on-Things (IoT) domain is not an exception, with more and more devices or sensors being connected to the Internet and transmitting different types of data. Usually, being part of more complex IT systems, it must be ensured that the IoT devices transmitting the data are authenticated components of the system before sending the data to a storage server. However, usually, IoT devices have limited computing power, therefore all of the work that they are doing should not be too expensive in terms of computations. This is the case for the authentication mechanism, too. Having this context, in this paper, we propose an authentication mechanism for IoT devices based on elliptic curves, which are known as having a low computational cost compared to other techniques used in cryptography that provide the same level of security. The proposed system includes a blockchain network that will verify the identity of the device which tries to connect within the system to send the data to the storage server, a process that will be made together with the storage server. Once the identity is valid, the blockchain records the transaction and the storage server initiates the data transmission process. Besides including a lightweight authentication mechanism, the proposed method has several other important properties due to it using the blockchain network. Compared to the related work that we analyzed, we show that the proposed authentication mechanism is secure against common attacks designed for IoT devices. The performance analysis shows that the authentication query made by the IoT device takes place in less than a second on both a MSP430F1611 microcontroller and a MICAz sensor.

## 1. Introduction

The last two years have been marked by important digital changes due to the fact that many activities have moved online, whether we are talking about human activities or activities based on electronic devices. Due to the COVID-19 pandemic, digital transformation has been accelerated in many areas; sometimes, activities that at first implied human factors have been rethought as automated activities, involving electronic elements such as devices or sensors. Therefore, with the rise of digitalization as an entirety, the Internet of Things (IoT) field has also evolved, although there have been situations, especially in 2020, when IoT projects may have been delayed due to the electronic components crisis generated by the pandemic. There have been several reports that analyze the IoT’s adoption and market which show considerable growth. For example, [1] shows that the IoT market was worth USD190.26 billion in revenue in 2021 and predicts the value to grow by 20% for 2023. On the other hand, the report [2] shows that in 2021 there were 12.2 billion active IoT devices and predicts a figure of 27 billion active IoT devices for 2025. However, why is the IoT industry so important? This is mainly due to the fact that, by using IoT devices, different types of data are collected, a fact that helps to monitor and manage remote systems in real-time. Moreover, the data itself is important because, through proper analysis, it can give useful insights about the system, making the business more efficient and productive. For example, with IoT devices, a patient can be monitored remotely or an intelligent home can be managed, manufacturing systems can be managed remotely, or vehicles can be tracked and controlled remotely.

However, there are several challenges in collecting data from IoT devices. One of the most important challenges is the security of the data. IoT devices may generate sensitive data, such as patient health data, data about manufacturing processes or procedures, or data generated by intelligent devices (voice assistants, cameras, etc.) related to human activities and discussions. The concern comes from the fact that IoT devices are mainly configured to be accessed from public Internet networks, and the data are sent to storage servers and accessed through web platforms or mobile applications. Another challenge, which is related to security, is data privacy. Here it is important that data is encrypted before being transmitted over the communication channel; in addition, it should be established who can access the data through powerful and well-defined access management. Another challenge for IoT systems is the volume and complexity of the data. There are situations in which IoT devices have to transmit a large amount of data or complex data with limited resources (for example, a limited bandwidth). In general, IoT devices generate and transmit unstructured data, for which big data analysis techniques should be applied in order to obtain relevant insights. 

An important aspect regarding the security of a system that embraces IoT devices is the IoT devices themselves, as they can be a sensitive component of the system. Being accessed from public networks makes them a very good target for attackers, who can use them as an entry point within the system or inject them with malware or ransomware that can further be spread across the entire system. In [3], the authors carried out a comprehensive study regarding IoT security from more perspectives, showing that attacks can occur on all three layers of IoT systems (application, network, and physical). Works [4,5,6,7,8,9] present comprehensive studies related to the security of the IoT domain, with different focuses: general threats, security solutions based on machine learning, or artificial intelligence. IoT devices are significant in industry, which has resulted in a specialized branch, Industry IoT (IIoT). Study [10] is focused on IIoT challenges and security solutions and proposes a four-layer architecture for IIoT systems. 

In any system, authentication and authorization are very important elements. IoT systems are no exception to this rule. Authentication is the process by which an authorized entity obtains access to a system, based on certain features or information, while authorization is the process by which an entity authenticated in a system receives consent to access certain resources or perform certain actions within the system based on the characteristics of the entity. In addition, related to authentication and authorization, the identities of the elements within the system should be managed properly. References [11,12,13] are studies related to identity in IoT applications. 

Another technology that has evolved in the last few years is blockchain, a type of distributed ledger technology. Although the initial application of blockchain was as a cryptocurrency [14], today blockchain has applications in many areas: finance, healthcare, supply chains, copyright protection, electronic voting systems, notary, and even cryptography for enhancing security, and so on. Recent studies have used blockchain technology within IoT systems for different purposes: management of IoT devices [15,16,17] or security [18,19,20].

Taking into consideration the actual technological context, we need to pay more and more attention to the way in which we secure our IT systems. To be sure that our systems are accessed only by authorized entities, even if we are talking about the users within the organization, e.g., clients and/or partners, or the smart devices connected to the system, a strong authentication mechanism is required. When we are talking about human factors, a strong authentication mechanism may include biometric features, but when we are talking about device authentication, we need to find other ways to authenticate them. IoT devices usually have a small amount of computational power; therefore, for them, the authentication mechanism should be strong but also lightweight such that it does not consume too much computational power. 

### 1.1. Contributions

Motivated by the context presented above, in this paper, we propose an authentication mechanism for IoT devices based on elliptic curves and bilinear pairing applied to elliptic curves. The reason why we used elliptic curves is that in general, the algorithms based on them are faster than other techniques that provide the same level of security. Work [21] has proven that elliptic curve cryptography (ECC) is faster than RSA (Rivest–Shamir–Adleman), with it providing the same level of security. The hardness assumption on which ECC is based is the discrete logarithm problem for elliptic curves. With them being faster, elliptic curves can be lightweight methods used for cryptographic purposes for IoT devices, taking into consideration that they have limited resources. To gain a higher level of degree for the authentication process, we include a blockchain network in the system which is used to validate the authentication. As a short description, when an IoT device is programmed to transmit data to a storage server, it should first authenticate using its identity. The device sends this request to the blockchain network that stores an identity structure as a smart contract, which will further communicate with the storage server in order to complete the validation process. If the IoT sends a valid identity, then the blockchain records this transaction, and the storage server contacts the IoT device in order to initiate the data transmission process. In summary, the contributions of the paper are as follows:We propose an authentication mechanism based on elliptic curves designed for IoT devices.We prove that our proposed authentication mechanism is secure against selective identity chosen-plaintext attack, and reply attack and provides user anonymity and unlinkability, while the analyzed related work do not provide these characteristics at the same time.In addition, using a blockchain network, it preserves the blockchain properties for query authentication, from which we can mention immutability, decentralization, and enhanced security because the characteristics of the network cannot be changed for the benefit of a particular entity.We show that the proposed authentication mechanism has good performance for IoT devices. Compared to the analyzed related work, the simulations show that we do not always achieve the best performance, but in these cases, the degree of security is higher.

### 1.2. Structure 

The rest of the paper is organized as follows: in the second section, we present background information about existing approaches for the authentication of IoT devices. Then, we present the main information about the mathematical tools that we will use: elliptic curves and bilinear pairing for elliptic curves. The third section contains the proposed authentication mechanism, in which we will describe the actors within the system and the algorithms for the proposed authentication, and we will discuss the security of the proposed method. In the fourth section, we discuss the implementation details, the results, and the performance. Finally, in the last section, we will present the conclusions of the paper.

## 2. Related Works

Authentication of the IoT devices within a system is a challenging task, therefore different approaches are proposed. For example, in [22], the authors propose TACACS, an authentication mechanism based on certificates. In addition, the authors prove that their proposed mechanism is secure against different types of attacks and efficient. In [23], the authors involve the physical layer of the IoT system within the authentication process and the process of authentication involves hyperelliptic curves. In [24], the authors propose two-factor authentication within an IoT system based on elliptic curves and they prove that the system is secure against sensor node captured attack. The authors of [25] propose an interesting approach for authentication; in their system, a “safe” area is defined in the proximity of a central point. If an IoT device that is outside that area tries to connect to the system, it is then rejected. However, if a legitimate device is outside the safe area, it cannot be authenticated.

Two relevant resources related to IoT are [26,27]. The first one presents technologies that can be used nowadays in IoT-based systems, such as artificial intelligence, blockchain, and game theory, and it also describes the challenges and potential solutions that may occur in the process of designing such systems. On the other hand, study [27] focuses only on the use of blockchain in IoT-based systems, with it presenting the challenges that can arise in developing such systems. Another interesting approach is given in [28], in which the devices involved within the system are authenticated by using X.509 certificates, but this work does not provide security analysis. In [29], the authors propose an authentication mechanism based on temporal credentials, random nonces, and IDs of the devices that change dynamically, but it does not contain security measures proposed for heterogeneous environments. Works [30,31] are authentication mechanisms based on three factors that use elliptic curves, elliptic curves, and fuzzy extractors, respectively. The limitations of [30,31] are related to the communications in which the IoT device is involved. In [32], an anonymous authentication method based on blockchain and elliptic curves is presented for distributed vehicle systems, with it making use of the on-board unit of the vehicle. On the other hand, study [33] proposes a security model for smart vehicles that uses access rights.

From a security perspective, the analyzed related work, ref. [25,28,29,30,31], do not provide at the same time security against selective identity chosen-plaintext attack, and reply attack, anonymity, and unlinkability.

## 3. Background Information

In this section, we provide the background information for elliptic curves and bi-linear maps for elliptic curves that we will use in the proposed system. Before going further, in Table 1, we present the notations used in this paper.

### 3.1. Elliptic Curves

The information in this section is synthesized from references [34,35], which are great resources for elliptic curve cryptography. When working with elliptic curves, the first thing needed is a finite field *G*. An elliptic curve defined over field *G* is given by the following equation (Equation (1)):*y*^2^ = x^3^ + *ax* + *b*,(1)
where *a* and *b* are elements from *G* that must fulfill the condition 4*a*^3^ + 27*b* ≠ 0 in order to avoid multiple roots. An interesting characteristic of elliptic curves is that they will lead to a group structure. For this, an identity (neutral) element should be given. This is a special point on the curve called the “point at infinity” (∞). Having this point, an elliptic curve defined over the field *G* is given by the following equation (Equation (2)): (2)$$E(G)=\{\infty \}\;\cup \;\{(xy)\;\in \;G\;\times \;G\;|\;y^{2}\;+\;x^{3}\;+\;ax\;+\;b\},$$

The elements within the set are called *points of the curve*. Furthermore, to accomplish a group structure, an operation between the points is needed. Therefore, over *E*(*G*) is defined as an additional operation between the points (more about how the addition is carried out between the points and its properties can be found in [34,35]), such that the structure (*E*(*G*), +) is a group whose identity element is the point at infinity. 

Elliptic curves make up an important branch of cryptography, which is based on the *discrete logarithm problem* (DLP) for elliptic curves. It states the following: given two points P and Q on a curve, it is hard to find the integer value *k* for which *Q = kP*. In this equality, the operation is point multiplication (*P*) with a scalar value (*k*). However, the curve must fulfill some conditions in order to be used in cryptosystems [34]; the field on which the curve is defined should have the form of $F_{p^{k}}$, large enough for the DLP to be hard. The number *p* is a large prime integer and the number *k* is the embedding degree of the curve. In addition, the order *n* of a base point *P* of the group should be large enough such that the system is secure against the Pollard attack for the DLP on the group <*P*>. In [34,35], some examples of settings for *n* are given. For example, the minimum number of bits for *n* should be 384 and for $p^{k}$ it should be 8194 in order to achieve a 192-bit security level. 

### 3.2. Symmetric Bilinear Pairing for Elliptic Curves

Symmetric bilinear pairings are maps between two structures as follows: let (*G*, +) and (*H*, ∙) be cyclic groups of order *q.* A symmetric bilinear pairing *e* is defined as e : G×G → H and has the properties of bilinearity (Equation (3)), nondegeneracy (Equation (4)), and computability (Equation (5)).
(3)$$\mathrm{For}\;\mathrm{all}\;a,b\;\in F_{q}^{*}\;\mathrm{and}\;\mathrm{for}\;\mathrm{all}\;P,Q\in G:\;e(aP,bQ)\;=\;e{\left(P,Q\right)}^{ab}$$
(4)e(P,P) ≠ 1, where 1∈H is the identity of H
(5)For all P,Q∈G, there is a polynomial-time algorithm A() for which e is easily computable

Examples of bilinear maps for elliptic curves are Weil or Tate pairing. Work [36] provides a comprehensive description of them.

## 4. Proposed Authentication Mechanism

In this section, we present the proposed system and describe its components, workflow, and security. In the initial stage of development, the owner of the system should specify the conditions based on which the IoT devices may access the blockchain network. All of these conditions are then implemented within a smart contract that will allow IoT devices to find the right blockchain node in the authentication process. 

### 4.1. Components of the System

The proposed system has several components, for which the characteristics are well-established:**Trusted authority (TA)**. This is a trusted entity that generates the keys for the rest of the components within the system.**Owner(s) of the system (*O_S_*)**. This is the entity to which the system belongs with all the physical and software elements. The owner of the system establishes which ones are the types of users within the system, the access policies, and the configurations of the IoT devices.**Users (U_S_)**. They are the entities who access the system, through a web/mobile/desktop application or platform. In order to access the system, a user needs to be authenticated, therefore the first authentication should be preceded by a registration process. According to the type of account, the user may access different parts of the application or platform.**Servers (S)**. The system can include servers for different purposes, but at least one server should be used for data reception from the IoT devices. This server has additional tasks, besides data reception; it collaborates with the blockchain network in order to validate an IoT device that requests to connect to the network and if the IoT device is authorized, then the server connects it to initiate data transmission. Other types of servers can be servers for applications/platforms or servers for different types of services involved within the system, but these do not affect the authentication process.**IoT devices (D_IoT_)**. These are the devices or sensors connected to a network and which send data to the storage server. In order to send the data, the IoT devices should be first authenticated within the system. For this, its identity should match a whitelist of IoT identities allowed to connect within the system.**Blockchain network (BN)**. Its purpose is to store the whitelist of IoT identities allowed to connect within the system. For authorizing an IoT device to connect to the system, the blockchain network collaborates with the storage server.

From the above description, there is an obvious delimitation between all the components of the system. This aspect is important in the design of a system, including for security reasons. For example, if one of the components is affected, then it should be easily replaced or isolated from the rest of the system until the problem is fixed. 

### 4.2. Architecture of the System

Figure 1 shows the architecture of the proposed system, including the components presented in the previous section and the interaction between them.

The components of the system are the trusted authority, owner, data users, IoT devices, blockchain network, and storage server. Firstly, the trusted authority generates a pair of public and private keys for the owner(s), data users, IoT devices, and storage server. Before being included within the system, the secret key for each IoT device is stored on it. Further, the data owner deploys the smart contract that allows IoT devices to find the right node within the blockchain network, when it will send the query authentication. In addition, the data owner deploys the smart contract used for the authentication process. Once the components are set up, IoT devices may send query authentication for the blockchain network to send further data. In order for the IoT device to find the right node, it sends a transaction to the smart contract address. When the right node is found, the IoT device sends an authentication request to the BN node. The BN node runs the smart contract deployed for the authentication process, by computing a validation value, which is sent to S, and S computes a corresponding value, which is sent back to BN. Furthermore, the BN node continues the computations from the authentication smart contract. If it succeeds, then the IoT device is allowed to authenticate within the system and the authentication transaction is validated. If the computations fail, the transaction is rejected, and the device cannot authenticate. The next section presents in detail the algorithms for the proposed system.

### 4.3. Algorithms for the Authentication Process

Authentication is one of the most important processes regarding a system because by being authenticated, an entity can access modules or resources of the system. From the user’s perspective, one of the most secure methods of authentication is biometric authentication, in which a user is asked to provide his/her unique biometric features (such as fingerprints, iris scanning, or facial recognition). However, not only humans may access a system. In our proposed system, IoT devices are other elements that try to access the system; therefore, they should be first authenticated. The proposed authentication scheme consists of several algorithms: setup, key generation, generation of the data structure which stores the identities of the IoT devices, an authentication query launched by the IoT device, and authorization for the authentication query. Not all algorithms should be called whenever an IoT device tries to connect. The setup algorithm, key generation, and generation of the data structure are called at an initial state of the system, and then every time when needed. The only algorithms called every time an IoT device tries to connect to the system are the *authentication query* algorithm and the *validation* algorithm. However, before the authentication request, the IoT device should find the right node of the blockchain network based on the smart contract deployed by the data owner. The authentication process works as follows: the IoT device submits a query authentication to the blockchain network. When received, the blockchain network works with the server in order to decide if the query authentication is valid. Therefore, the blockchain network computes a specific value and sends it to the server. Furthermore, the server interrogates the structure of stored identities and sends the validation value back to the blockchain network. Here, the blockchain network uses the validation value and checks with the smart contract whether the validation query is valid. If the validation query is valid, then the IoT device is authenticated within the system and can send data, otherwise, the authentication query is rejected. 

Furthermore, we present the algorithms involved in the authentication process:

**1. Setup**$\left(\lambda \right)\to PS$. The setup algorithm is called by the TA. The input is the security parameter λ and the outputs are the parameters of the system $PS=\left(E(F_{p^{k}}\right),\;H,\;h_{1})$, which contain the following elements:
The elliptic curve $E(F_{p^{k}})$, which is defined over $F_{p^{k}}$ as it is described in [34]: *p* is a large prime integer and the number *k* is the embedding degree of the curve;The generator point H of the group $\left(E(F_{p^{k}}),+\right)$ whose order is *n*, with *n* having the properties described in [34];The hash function $h_{1}$ that will be used in further algorithms.


**2. KeyGeneration**$\left(PS\right)\to \left(\left(p_{O_{S}},\;s_{O_{S}}\right),\;\left(p_{D_{IoT}},\;s_{D_{IoT}}\right),\;\left(p_{S},\;s_{S}\right)\right)$. The key generation algorithm is also called by the TA, taking as input the parameters of the system PS and returning as output the pairs of public and secret keys for the *O_S_*, *D_IoT_*, and *S*. The keys are generated similarly, following the procedure below:
Owner of the system: randomly generate an integer $w\leq \#E\left(F_{p^{k}}\right)$ (where $\#E\left(F_{p^{k}}\right)$ is the notation for the number of points on the curve) and achieve the point using Equation (6):(6)W=wH 


The owner’s public key is $p_{O_{S}}=W$ and the secret key is $s_{O_{S}}=w$.


IoT device: randomly generate an integer $d\leq \#E\left(F_{p^{k}}\right)$ and achieve the point using Equation (7): (7)D=dH 


The public key is computed using Equation (8).
(8)$$p_{D_{IoT}}=\left(p_{D_{IoT}1,\;}p_{D_{IoT}2}\right)=\left(e{\left(H,H\right)}^{\frac{1}{d}},\;D\right)$$
and the secret key is $s_{D_{IoT}}=d$.


Server: randomly generate an integer $s\leq \#E\left(F_{p^{k}}\right)$ and achieve the point using Equation (9): (9)S=sH 


The server’s public key is $p_{S}=S$ and the secret key is $s_{S}=s$. 

**3. GenerateIdentitiesStructure**$\left(PS,\;p_{O_{S}}\right)\to DS$. The algorithm for generating the structure of identities is called by the owner of the system. The purpose of this algorithm is to store in a data structure all of the identities for the IoT devices that are allowed to access the system. The data owner should make the following steps for each valid identity i in order to be added to the structure:
*O_S_* randomly generates the value $v\in F_{p^{k}}$ and then computes the values Equations (10)–(13):(10)$$DS_{i0}=e{\left(H,H\right)}^{v}\;$$
(11)$$DS_{i1}=v\cdot p_{D_{IoT}2}=vD=vdH\;$$
(12)$$DS_{i2}=e{\left(H,H\right)}^{wv}\;$$
(13)$$DS_{ij}=vh_{1}\left(i\right)H\;$$All of the elements of the form ($DS_{i0},\;DS_{i1},\;DS_{i2},\;\mathrm{and}\;DS_{ij}$) are used to generate a data structure *DS*. Note that this data structure hides the valid identities under the hash function $h_{1}$. Before being connected to the network, the IoT device is configured to store an encrypted version of the value *v*. To encrypt the value v, the owner of the system may choose any encryption system that is suitable for the IoT device. The resulting data structure *DS* is then registered to the blockchain network, as a smart contract. 


**4. AuthenticationQuery**($p_{S},p_{D_{IoT}},I,\;s_{D_{IoT}})\to Q$. The authentication query algorithm is called by the IoT device when it tries to authenticate itself within the server. In order to find the right node within the blockchain network, the IoT device should meet the conditions implemented in the smart contract defined by the data owner. The input for the authentication query algorithm consists of the public key of the server, the public key of the IoT device, the identity of the IoT device, and the private key of the device. The output is a query value that will be further sent to the blockchain network. To compute the query value Q, the device proceeds as follows:Randomly generate the value $m\in F_{p^{k}}$ and set $Q_{1}=m$.Calculate the value $Q_{2}$ using Equation (14):(14)$$Q_{2}=v\left[\left(-m\cdot p_{S}+p_{O_{S}}\right)-\left(s_{D_{IoT}}-h_{1}\left(I\right)\right)\cdot H\right]=v\left[\left(-m\cdot s\cdot H+wH\right)-\left(d-h_{1}\left(I\right)\right)H\right]=v\left(-ms+w-d+h_{1}\left(I\right)\right)H$$The query authentication is the pair $Q=\left(Q_{1},Q_{2}\right)$.

**5. Validation**$\left(DS,Q,s_{S}\right)\to b$**.** Through the validation algorithm, it is decided whether the IoT device has permission to authenticate within the system or not. The algorithm is performed by the blockchain network node to which the authentication query was sent in collaboration with the server. This algorithm takes as an input the data structure of the identities, the query received from the IoT device, and the secret key of the server. The output is a Boolean value, true meaning that the IoT device can be authenticated and false meaning that the IoT device is not authorized to authenticate. For validation, the following steps are performed:The value $A_{2}^{\prime }=DS_{i0}^{Q_{1}}$ is computed by the blockchain node and it is sent to the server.The value $A_{2}$ is computed using Equation (15):(15)$$A_{2}={\left({A^{\prime }}_{2}\;\right)}^{s_{S}}={\left(DS_{i0}^{Q_{1}}\right)}^{s_{S}}={\left({\left(e{\left(H,H\right)}^{v}\right)}^{m}\right)}^{s}=e{\left(H,H\right)}^{vms}\;$$$A_{2}$ is computed by the server, which sends this value to the blockchain node.



The value $A_{1}$ is computed as follows within the blockchain node based on Equation (16):(16)$$A_{1}=e\left(H,Q_{2}\right)\cdot e\left(H,DS_{i1}-DS_{i\sigma \left(k\right)}\right)\;$$Here, σ is an indexing function that should generate the *j* value $j=\sigma \left(k\right)$. Another important value is $A_{3}$, which is also computed within the blockchain node using Equation (17): (17)$$A_{3}=DS_{i2}=e{\left(H,H\right)}^{wv}$$


If the equality $A_{3}=A_{1}\cdot A_{2}$ takes place, then the blockchain network node validates and records the transaction. This means that the identity of the IoT device is correct; therefore, it can authenticate within the system.


Furthermore, the server may contact the IoT device with identity $\sigma \left(k\right)$ to initiate data transmission.Figure 2 shows the authentication process. In *authenticationQuery,* the input values ps, pdiot, I, sdiot represent $p_{S},p_{D_{IoT}},I,\;s_{D_{IoT}}$ from the query authentication algorithm; in *askValidationValue*, ds_i0_q1 represents $DS_{i0}^{Q_{1}}$; in *computeValidationValue*, a2ss represents ${\left({A^{\prime }}_{2}\;\right)}^{s_{S}}$; in *sendValidationValue*, a2 represents $A_{2}$; in *checkAuthenticationValidation*, A1, A2, A3 represent $A_{1},\;A_{2},\;A_{3}$; in *registerAuthenticationTransaction*, *authenticationFailed*, and *rejectAuthenticationTransaction*, Q represents the query authentication.From the proposed scheme, the authentication query and validation are the most used because each time an IoT device wants to connect to the system, it sends an authentication query. However, all of the algorithms work with operations on elliptic curves, more specifically scalar multiplications of points on the curve. In [34], it was shown that this operation has the complexity of $O\left(\log_{2}k\right)$, where k is the scalar with which the point is multiplied. Because the setup and key generations algorithms work only with operations on points, their time complexity is $O\left(\log_{2}n\right)$. Another aspect that influences the complexity is the bilinear pairing used in the algorithms. In [36], it was is shown that Tate pairing for elliptic curves has a better performance than Weil paring, but the curve should be chosen more carefully. Therefore, besides operations on points, the time complexity for the rest of the algorithms, namely identity structure generation, query authentication, and validation, depends on the hash function used and the bilinear pairing used. This leads to a time complexity defined as $\max\left\{O\left(\log_{2}n\right),\;T\left(h\right),\;T\left(bp\right)\right\},$ where $T\left(h\right)$ represents time complexity of the hash function and $T\left(bp\right)$ represents the time complexity for the bilinear pairing.


## 5. Security Analysis

In this section, we discuss the correctness of the proposed authentication mechanism and its security. 

### 5.1. Correctness

For any proposed scheme, it is important to discuss its correctness; therefore, in this subsection, we prove that the proposed authentication mechanism is correct. Its correctness is given by the verification $A_{3}=A_{1}\cdot A_{2}$ (Equation (18)), which shows that if an identity is valid, then it is for sure found within the data structure of identities. Indeed, using the properties of the bilinear maps described in Section 3.2, the equality from the verification occurs every time when the blockchain network receives an authentication request from an IoT device with a valid identity. In addition, all of the coefficients and helper functions are chosen in $F_{p^{k}}$; therefore, the operations with points on the curve yield to values that remain in the same group generated by the elliptic curve.
(18)$$\begin{array}{cc} A_{1}\cdot A_{2} & =\left(e\left(H,Q_{2}\right)\cdot e\left(H,DS_{i1}-DS_{i\sigma \left(k\right)}\right)\right)\cdot e{\left(H,H\right)}^{vms}\\ & =e\left(H,\;v\left(-ms+w-d+h_{1}\left(I\right)\right)H\right)\cdot e\left(H,\;vdH-vh_{1}\left(i\right)H\right)\cdot e{\left(H,H\right)}^{vms}\\ & =e{\left(H,H\right)}^{v\left(-ms+w-d+h_{1}\left(I\right)\right)+vd-vh_{1}\left(i\right)+vms}=e{\left(H,H\right)}^{vw}=A_{3}\; \end{array}$$

The security model consists of the participants described in Section 4.1 and the algorithms and authentication query and validation protocols described in Section 4.3. After running the authentication query and validation, an IoT device can be in one of the following states: accepted (meaning the authentication is approved) or rejected (meaning the authentication is rejected). In addition, the security model involves an adversary that has access to all public elements. More about the adversary will be discussed in Section 5.2. 

Because the proposed authentication scheme involves a blockchain network in the validation process, it has the following properties: decentralization of the validation process, persistency of valid authentication queries, the anonymity of the IoT device that requests authentication, and audibility of the logged transactions using the timestamps within the blockchain network.

### 5.2. Security

In this section, we show that the proposed scheme is secure against selective identity chosen-plaintext attack, and reply attack, too. 

*Formal analysis using BAN logic*. After the IoT device finds the right blockchain node based on the smart contract, it can send authentication requests. For query authentication (Q), $D_{IoT}$ initiates an authentication request for the blockchain network’s node (BNN). We will prove that $D_{IoT}$ believes Q, and the blockchain network node (BNN) should trust the IoT device in order to accept and record the request authentication transaction, meaning that we should also obtain BNN |≡Q (BNN believes the query authentication Q sent by $D_{IoT}$). This is shown below.

$D_{IoT}|\equiv Q$. This means that $D_{IoT}$ believes Q as it is computed by itself and due to the following rules: $D_{IoT}|\equiv p_{S}$ and $D_{IoT}|\equiv p_{O_{S}}$ and $D_{IoT}|\equiv s_{D_{IoT}}\;$because $TA|\Rightarrow p_{S}$ and $TA|\Rightarrow p_{O_{S}}$ and $TA|\Rightarrow s_{D_{IoT}}$, meaning that $D_{IoT}$ believes $p_{S},\;p_{O_{S}},\;s_{D_{IoT}}$, because TA has jurisdiction over them (Equations (6)–(9)).

BNN|≡Q. This means that BNN believes the query authentication sent by the $D_{IoT}$. This is due to the following logic: #m (m used to generate Q is fresh from Equation (14)) and $BNN\left|\equiv Q\right|\sim m$ (BNN believes Q uttered m). Furthermore, $S\left|\equiv A_{2}^{\prime }\right|\Rightarrow BNN$ (S believes $A_{2}^{\prime }$ because BNN has jurisdiction over it), $BNN|\equiv A_{2}$, because $A_{2}^{\prime }$ involved in the computation of $A_{2}\;$was generated by BNN itself and $TA|\Rightarrow s_{S}$ (TA has jurisdiction over $s_{S}$) (Equation (15)). Furthermore, $BNN|\equiv A_{1}$ and $BNN|\equiv A_{3}$ because $A_{1}\;\mathrm{and}\;A_{3}$ are computed by themselves and $O_{s}|\Rightarrow DS$ ($O_{S}$ has jurisdiction over DS used in Equations (16) and (17). Because $BNN|\equiv A_{1}$ and $BNN|\equiv A_{2}$ and $BNN|\equiv A_{3}$ (BNN believes $A_{1},\;A_{2},\;A_{3}$), it can check that $A_{3}=A_{1}\cdot A_{2}$. Indeed, if this equality takes place, then the authentication query transaction is recorded and the IoT device is authenticated.

*Security against selective identity chosen-plaintext attack*. In order to prove this security, we will use a game between a challenger (C) and an adversary (A). The scheme is secure against this type of attack if the advantage that the adversary has is negligible regarding the security parameter λ. Here, the adversary is any probabilistic polynomial-time (PPT) algorithm. Therefore, the game works as follows:The adversary $A\left(\lambda \right)$ chooses an identity value $I^{*}\in {\left\{0,1\right\}}^{*}$.The challenger C runs the setup algorithm, which outputs the parameters of the system, and the key generation algorithm, which outputs the tuple $\left(\left(p_{O_{S}},\;s_{O_{S}}\right),\;\left(p_{D_{IoT}},\;s_{D_{IoT}}\right),\;\left(p_{S},\;s_{S}\right)\right)$ as described above, and makes the public key available. The adversary may use any of the public keys.The adversary A may ask multiple times for the query authentication pair $Q=\left(Q_{1},Q_{2}\right)\leftarrow \mathrm{AuthenticationQuery}\left(p_{S},p_{D_{IoT}},I\right)$ for any identity I, except for $I^{*}$, and the challenger C sends the corresponding values.After an arbitrary number of queries, the adversary makes two messages $m_{0}$ and $m_{1}$ of arbitrary lengths available. The challenger randomly picks the value $b\in \left\{0,1\right\}$ and challenges the attacker with the value $C^{*}\leftarrow \mathrm{AuthenticationQuery}\left(p_{S},p_{D_{IoT}},m_{b}\right)$.Furthermore, the adversary may ask for additional query values, except for $I^{*}$.Finally, at an arbitrary point, the adversary outputs the value $b^{\prime }$. 

The adversary wins the game if the following equality takes place: $b=b^{\prime }$. From the above game, the adversary wins if it correctly guesses the values of the challenge authentication pair. Therefore, its advantage should be negligible in order for the proposed scheme to be secure against selective identity chosen-plaintext attacks. The adversary should actually discover the authentication query which consists of two components. Particularly, the adversary should find the value $m\in F_{p^{k}}$, a fact for which the probability of guessing the value for m is $\frac{1}{p^{k}}$. The advantage of the adversary is given by the formula below (Equation (19)):(19)$$Adv\left(A\right)=\frac{1}{p^{k}}\;$$

The advantage value is negligible; therefore, the proposed authentication scheme is secure against selective identity chosen-plaintext attacks. 

*Reply attack*. In reply attacks, the identity or information related to it is stored and used at a future moment. Even if an attacker captures a query authentication and saves it for later use, or even if the IoT device stores the current query authentication, it cannot be used for further authentication queries because every time the IoT device computes the query authentication, the value $m\in F_{p^{k}}$ is randomly generated.

*User anonymity and unlinkability*. The proposed authentication scheme provides anonymity. This is due to the fact that when the IoT device tries to connect to the system it needs to use its identity, indeed. However, the identity is not revealed in the output of the query authentication algorithm. Instead, the identity is hidden under the query Q composed of two components that involve a randomly generated number, and additionally, the identity is hashed during computations. Therefore, even if an attacker captures the query authentication, the identity of the IoT device will not be revealed. 

Furthermore, we compare the results in terms of security with other existing similar works. Table 2 shows a short description, and the limitations of the related work and Table 3 shows the comparison of related work with our proposed scheme.

### 5.3. Blockchain Enhancing Security

The validation of the authentication process is performed by the blockchain network; therefore, the server cannot learn anything about the identity of the IoT device that queries the authentication. In addition, the server is unable to generate a valid authentication query because in this process, it involves the secret key of the IoT device. Because the secret key of the IoT device is $d\in F_{p^{k}}$ this leads to the probability of the server guessing the value d as $\frac{1}{p^{k}}$, which is a probability value that is negligible.

Another property of the proposed authentication mechanism is the immutability resulting from the use of the blockchain. Once a transaction is recorded, it cannot be modified or deleted; therefore, historical authentication data can be reviewed anytime (auditing).

## 6. Performance

For the authentication query, the IoT device has to make the following operations: computing $Q_{1}$ by generating a random number and computing $Q_{2}$ which involves point multiplication with a value (this value is obtained by applying operations on integer values, from which one of them is randomly generated), as well as applying a hash function. Table 4 contains the description of operations used in our proposed authentication scheme made by the IoT device, as well as other operations used by the related work. The simulations from this section were made on a 64-bit Windows 10 laptop, with 16 GB RAM, Intel Core i3, 2.40 GHz. 

Note that $T_{ECC}$ is a generic operation between points on elliptic curves, without mentioning whether it is addition, fixed multiplication, or random multiplication. However, the operations do not have the same cost, with random multiplication being the costliest, followed by fixed multiplication and lastly, addition.

According to [37] by using a 169-bit secure elliptic curve implemented on a MSP430F1611 microcontroller, the operations on points require the amount of time described in Table 5. The specifications for this microcontroller are 10 kB RAM, 48 kB flash memory, and 8 MHz clock frequency [37]. In the proposed authentication mechanism, the IoT device should randomly generate the value $Q_{1}$ and then compute the value $Q_{2}$. The costliest operation for computing $Q_{2}$ is the multiplication operation with a randomly generated value; therefore, the time for the IoT device to make these computations is approximately 1.31 s and it is computed as follows:(20)$$T_{IoT}=T_{Rmult}=1.31s\;$$

This performance is achieved by simulations using an elliptic curve with a security level of 169 bits. Following the comparison in [37], the best score regarding the performance of our scheme can be achieved using an elliptic curve with a 159-bit level of security and the time of the authentication query achieved with this type of curve would be estimated at 0.48 s. Table 6 summarizes these results.

If we follow the simulations from [31] in which the authors used a MICAz sensor (4 kB RAM and 128 kB flash memory), the costliest operation is point multiplication on the elliptic curve, which requires 114 ms; therefore for our scheme, the total amount of time required for IoT authentication takes 117.63 ms on the MICAz sensor. Table 7 provides the comparison cost in terms of operations made by each related work and our work, while Table 8 shows the time for each operation applied on the MICAz sensor and Table 9 shows the performance comparison with related work in terms of milliseconds. Figure 3 shows these results graphically. An important remark is that in the related work, the IoT sensors are used to authenticate the users, not the devices themselves; therefore, in the related work, the operations made by IoT devices are not too complex. However, our proposed mechanism provides stronger security for the system by authenticating the IoT device itself.

### TLS Handshake

During the measurement process, we observed that the handshake packets were sometimes lost and the necessity to retransmit them was required, either by the client or by the server, with a slight increase in time for the execution process. As we can observe in Figure 4, we have an example that shows the transport layer security (TLS) handshake using the MICAz sensor. Since our focus is not dedicated to reliability, we have eliminated those outliers that are different from the median by 18 percent or even more. In Figure 5, we can see the corresponding adjustments for the measurements. The same removal method has been applied for all of the measurements of the handshake. Based on the sources mentioned and using the same analysis of the identical of benchmark for the cryptographic primitives, the TLS handshake benchmark was repeated 100 times. Table 10 shows the remaining quantity and number of valid measurements of the TLS handshake once the remove method has been applied for the outliers that were caused by the MICAz sensor.

## 7. Conclusions

IoT systems involve complex processes and sensitive data that should be protected against unauthorized access and different types of attacks. One of the most important processes within information systems, in general, is the authentication process, even if it is about humans or devices that try to connect to the system. In this paper, we proposed an authentication mechanism for IoT devices based on elliptic curves and blockchain. The motivation for using elliptic curves lies in the fact that they are more lightweight, which is more suitable for IoT devices with limited computing power than other techniques which give the same levels of security. We included in the proposed authentication mechanism a blockchain network that records all authentication queries based on the smart contract. Additionally, the blockchain communicates with a server in order to decide whether the authentication query is valid or not. We obtained good performance, with the authentication query made by the IoT device taking place in less than a second on both an MSP430F1611 microcontroller and a MICAz sensor. The strength of the proposed authentication mechanism is given by the properties of the blockchain but also by the security of the authentication mechanism. In comparison with the related work, it is not the fastest process, but provides stronger authentication for the system. As a further research direction, we want to pass the simulation phase and implement the proposed scheme in a real environment. The proposes authentication mechanism can be applied on a wide range of domains. In practical applications for different disciplines, such as, nano or microstructured sur-face, the security of data and information systems involving different sensors can be ensured for applications in computational physics, such as [38] or [39].

## Figures and Tables

**Figure 1 sensors-23-01371-f001:**
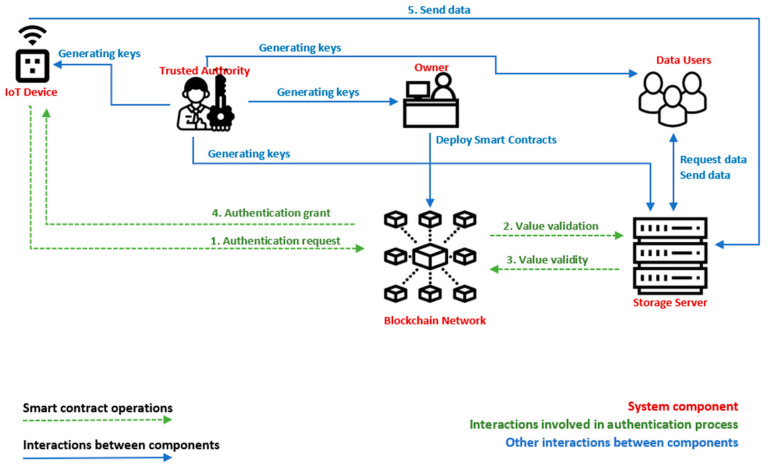
The architecture of the proposed system.

**Figure 2 sensors-23-01371-f002:**
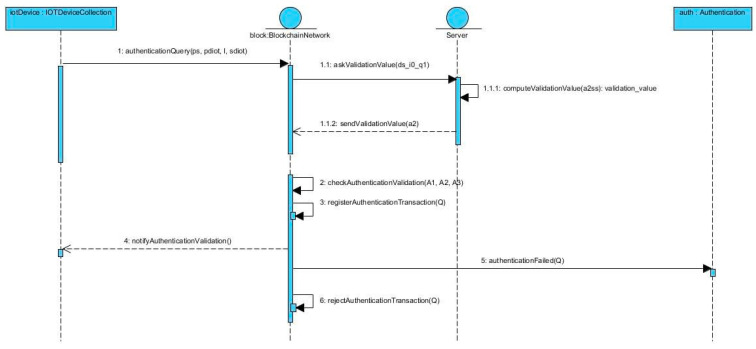
The authentication process.

**Figure 3 sensors-23-01371-f003:**
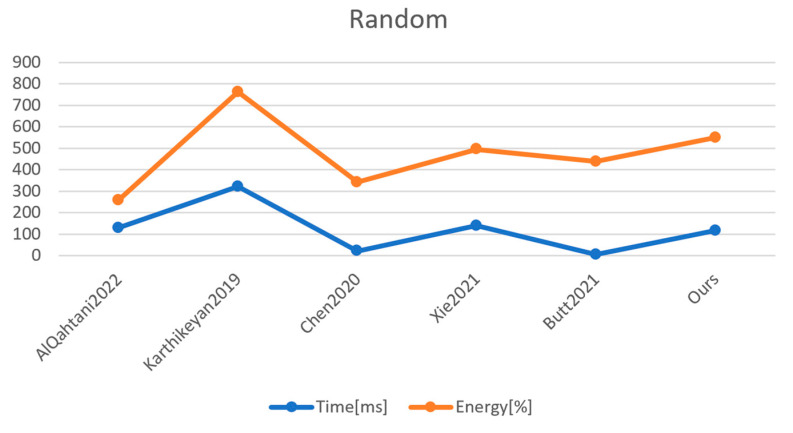
Time and energy measured for 32 bytes of random data that are relative within the implementation of the references mentioned. The median of the values, rounded to the nearest percent, and n = 100 are taken into consideration. References: AlQahtani2022 [25], Karthikeyan2019 [28], Chen2020 [29], Xie2021 [30], Butt2021 [31].

**Figure 4 sensors-23-01371-f004:**
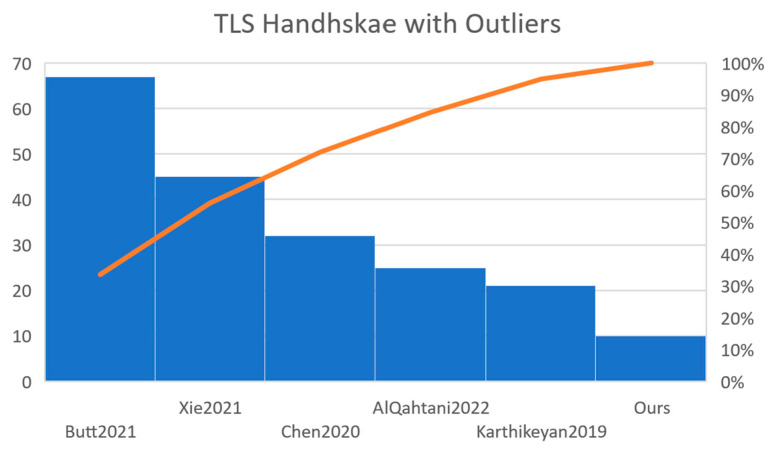
Execution time of the TLS handshake using the MICAz sensor. References: AlQahtani2022 [25], Karthikeyan2019 [28], Chen2020 [29], Xie2021 [30], Butt2021 [31].

**Figure 5 sensors-23-01371-f005:**
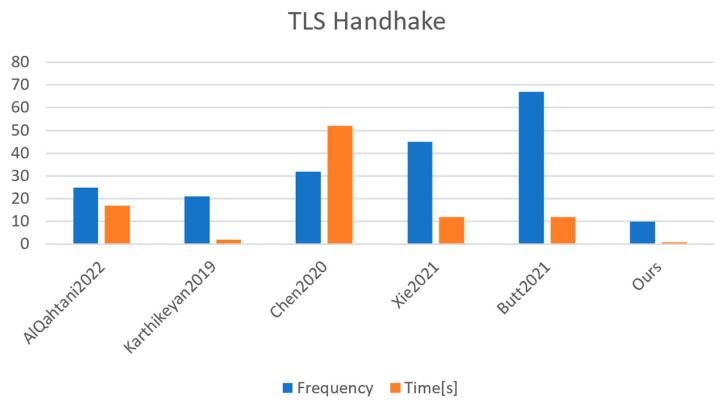
Execution time of the TLS handshake using the MICAz sensor without outliers and which has more than 25% from the median. References: AlQahtani2022 [25], Karthikeyan2019 [28], Chen2020 [29], Xie2021 [30], Butt2021 [31].

**Table 1 sensors-23-01371-t001:** Notations used.

Notation	Description
*G*	Finite field
$E\left(G\right)$	Elliptic curve defined over group G
$P,Q,\;H,\;Q_{2}$	Points on elliptic curves
$F_{p}$	Galois field of order p
*A()*	Polynomial-time algorithm
TA	Trusted authority (component of the proposed system)
$O_{S}$	Owner(s) of the system (component of the proposed system)
$U_{S}$	User of the system (component of the proposed system)
S	Server (component of the proposed system)
$D_{IoT}$	IoT device (component of the proposed system)
BN	Blockchain network (component of the proposed system)
λ	Security parameter
PS	Parameters of the system
$\left(p_{x},\;s_{x}\right)$	The pair of the public and secret keys, where x is the component of the system
$h_{1},h_{2}$	Hash functions
DS	Data structure of IoT devices’ identities
$e\left(\cdot ,\cdot \right)$	Bilinear pairing
σ	Indexing function

**Table 2 sensors-23-01371-t002:** Summary of related work.

Resource	Description	Limitations
[25]	In their system, a “safe” area is defined in the proximity of a central point. If an IoT device that is outside that area tries to connect to the system, then it is rejected.	If a legitimate device is outside the safe area, it cannot be authenticated.
[28]	The devices within the system are authenticated through X.509 certificates.	The security analysis is not provided.
[29]	The authentication mechanism uses temporal credentials and IDs that change dynamically.	There are no security measures proposed for heterogeneous environments.
[30]	Three-factor authentication mechanism based on elliptic curve cryptography.	The model of the network makes the communication between the user and IoT device direct and remote, without passing a gateway.
[31]	Three-factor authentication mechanism based on elliptic curve cryptography and fuzzy extractor.	The accuracy of the information exchanged between IoT devices is not guaranteed.

**Table 3 sensors-23-01371-t003:** Comparison of security properties.

Resource	Selective Identity Chosen-Plaintext Attack	Reply Attack	Anonymity	Unlinkability
[25]	✗	✗	✗	✓
[28]	✗	✗	✗	✗
[29]	✗	✓	✗	✗
[30]	✗	✓	✓	✓
[31]	✗	✓	✓	✗
Ours	✓	✓	✓	✓

✓—Has property/✗—not mentioned/does not have the property.

**Table 4 sensors-23-01371-t004:** Description of operations.

Operation	Description
$T_{Fmult}$	Time required to make a multiplication operation with a fixed number
$T_{Rmult}$	Time required to make a multiplication operation with a randomly generated number
$T_{add}$	Time required to make an addition operation between points
$T_{h}$	Time required to apply a hash function
$T_{ECC}$ [30]	Time required to make an ECC (elliptic curve cryptography) operation
$T_{ECDH}$ [31]	Time required for ECDH (elliptic curve Diffie–Hellman) key exchange

**Table 5 sensors-23-01371-t005:** The time required for operations on points of an elliptic curve according to [37] on the MSP430F1611 microcontroller.

Operation	Description	Value (s)
$T_{Fmult}$	Time required to make a multiplication operation with a fixed number	0.65
$T_{Rmult}$	Time required to make a multiplication operation with a randomly generated number	1.310
$T_{add}$	Time required to make an addition operation between points	0.00964

**Table 6 sensors-23-01371-t006:** Summary of authentication request for different levels of security on a MSP430F1611 microcontroller.

Security	Time (s)
169-bit	1.31
159-bit	0.48

**Table 7 sensors-23-01371-t007:** Comparison of operations made by the IoT device.

Resource	Operations
[25]	-
[28]	-
[29]	$6T_{h}$
[30]	$7T_{h}+T_{ECC}$
[31]	$T_{h}+T_{ECDH}$
Ours	$T_{Rmult}+T_{h}$

**Table 8 sensors-23-01371-t008:** The time required for operations on points of an elliptic curve according to [31] on the MICAz sensor.

Operation	Value (ms)
$T_{h}$ [31]	3.63
$T_{ECC}$ [30]	114
$T_{ECDH}$ [31]	2.375

**Table 9 sensors-23-01371-t009:** Comparison of operations made by the IoT device.

Resource	Operations	Time (ms)	Energy (%)
[25]	-	25	129
[28]	-	21	442
[29]	$6T_{h}$	21.75	321
[30]	$7T_{h}+T_{ECC}$	139.41	356
[31]	$T_{h}+T_{ECDH}$	6.005	432
Ours	$T_{Rmult}+T_{h}$	117.63	432

**Table 10 sensors-23-01371-t010:** Execution time of the TLS handshake using the MICAz sensor.

Resource	Operations (ms)	Frequency	Time (s)
[25]	-	25	17
[28]	-	21	2
[29]	$6T_{h}$	32	52
[30]	$7T_{h}+T_{ECC}$	45	12
[31]	$T_{h}+T_{ECDH}$	67	12
Ours	$T_{Rmult}+T_{h}$	10	1

## Data Availability

Not applicable.

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
