# Peer review of "Elliptic Curve-Based Query Authentication Protocol for IoT Devices Aided by Blockchain"

_sensors, 2023, doi:10.3390/s23031371_

Round 1

Reviewer 1 Report

Authors presented the proposed system that includes a blockchain network that will verify the identity of the device which tries to connect within the system to send the data to the storage server.

Abstract is too lengthy.

Equations across the paper should be numbered.

The presented work is just a simulation work. Authors should justify the significance of presenting this simulation research.

A separate section with proposed system architecture and explanations should be included.

Author Response

Dear Reviewer,

Thank you for your constructive comments. Please, see our answer for each comment below.

The authors presented the proposed system that includes a blockchain network that will verify the identity of the device which tries to connect within the system to send the data to the storage server.

Abstract is too lengthy.

Answer: We changed the abstract, by condensing the same information within more concise phrases but keeping the same ideas. Also, we highlighted the advantages of the proposed system.

Equations across the paper should be numbered.

Answer: Equations from section 3 were already numbered. However, we added numbers for all equations within sections 4 and 5.

The presented work is just a simulation work. Authors should justify the significance of presenting this simulation research.

Answer: We made this study and presented the results through the simulations motivated by the context presented in section 1 (we also stated in the conclusions section as further research that we will implement in a real environment the proposed scheme). Compared to related work, the simulations show that we achieve a good performance with a higher degree of security. However, we highlighted the strengths of the proposed authentication scheme in the abstract and section 1.1.

A separate section with proposed system architecture and explanations should be included.

Answer: we added section 4.2 which includes the system architecture and explanations.

Reviewer 2 Report

The authors proposed an authentication protocol for IoT devices by applying elliptic curves and blockchain.  Security analysis is conducted and overhead is evaluated.

Some issues with current version.

(1) 'blockchain' did not appear in the paper title. why?

(2) Pseudonym and blockchain have been applied in designing authentication, like 'BLA: Blockchain-Assisted Lightweight Anonymous Authentication for Distributed Vehicular Fog Services'. What is the difference from these existing works?

(3) BAN logic may be used for formal analysis of security.

Author Response

Dear Reviewer,

Thank you for your constructive comments. Please, see our answer for each comment below.

The authors proposed an authentication protocol for IoT devices by applying elliptic curves and blockchain.  Security analysis is conducted and overhead is evaluated.

Some issues with current version.

(1) 'blockchain' did not appear in the paper title. why?

Answer: The focus is on the operations made by the IoT device, which are operations on elliptic curves. However, we slightly changed the title in order to include “blockchain”.

(2) Pseudonym and blockchain have been applied in designing authentication, like 'BLA: Blockchain-Assisted Lightweight Anonymous Authentication for Distributed Vehicular Fog Services'. What is the difference from these existing works?

Answer: We added a discussion about this study (reference [36]) at the end of section 2.

(3) BAN logic may be used for formal analysis of security.

Answer: We added in section 5.2 formal analysis using BAN logic.

Reviewer 3 Report

Oriented to the security of the IoT scenarios, the authors propose an authentication mechanism for IoT devices based on elliptic curves and a blockchain network that will verify the identity of the device. Results show that the proposed authentication mechanism is secure against common attacks designed for IoT devices. The topic is interesting and the paper is well organized. However, there are several issues need to be addressed furthermore. Please find my detailed comments as follows:

1) The abstract needs to be improved significantly. From the abstract, it is better to summarize the limitations of existing works to highlight the necessity of the study given that this topic is not new. From the abstract, I can not find any novel points about the solution considering that both elliptic curves and blockchain networks have been widely investigated. This is no description of how the verify the effectiveness of the proposed solution. Finally, no concrete experiment results demonstrate the advantages of the proposed solution.  

2) In section 1, several related works are listed, but there is no discussion about the limitations or weak points of the existing works. Please highlight the necessity of the study. Both elliptic curves and blockchain networks are not new anymore, maybe your contribution is in the combination of these two prior arts, but it is not easy to see the novel points.

3) In section 2, it can be found that many studies oriented to IoT security are based on elliptic curves, what’s the problem with the existing solutions? Besides, there should be numerous IoT security-related works based on blockchain, please discuss related typical prior arts

4) In 4.1, for the high scalability of the authentication framework, the blockchain should be a distributed system. However, what is the relationship between each blockchain entity and IoT devices, and how does each IoT device find the right blockchain node? In Figure 1, the blockchain looks like a single node, but it is not true.

5) In 4.2, please highlight what’s the difference between the proposed algorithm and the ECC. In addition, it is better to make a complexity analysis of the proposed algorithm

6) In section 6, please the detailed simulation environment setting in case the performance of different approaches is evaluated by simulations.

Author Response

Dear Reviewer,

Thank you for your constructive comments. Please, see our answer for each comment below.

Oriented to the security of the IoT scenarios, the authors propose an authentication mechanism for IoT devices based on elliptic curves and a blockchain network that will verify the identity of the device. Results show that the proposed authentication mechanism is secure against common attacks designed for IoT devices. The topic is interesting and the paper is well organized. However, there are several issues need to be addressed furthermore. Please find my detailed comments as follows:

1) The abstract needs to be improved significantly. From the abstract, it is better to summarize the limitations of existing works to highlight the necessity of the study given that this topic is not new. From the abstract, I can not find any novel points about the solution considering that both elliptic curves and blockchain networks have been widely investigated. This is no description of how the verify the effectiveness of the proposed solution. Finally, no concrete experiment results demonstrate the advantages of the proposed solution.  

Answer: We changed the abstract, by condensing the same information within more concise phrases but keeping the same ideas. Also, we highlighted the strengths of the proposed authentication scheme at the end of it.

2) In section 1, several related works are listed, but there is no discussion about the limitations or weak points of the existing works. Please highlight the necessity of the study. Both elliptic curves and blockchain networks are not new anymore, maybe your contribution is in the combination of these two prior arts, but it is not easy to see the novel points.

Answer: References from section 1 are used to show the context (the advance of the IoT systems) and the motivation (limited computing power of IoTs) for the proposed authentication scheme. We made this study and presented the results through the simulations motivated by the context presented in section 1 (we also stated in the conclusions section as further research that we will implement in a real environment the proposed scheme). Compared to related work, the simulations show that we achieve a good performance with a higher degree of security. We modified section 1.1 and highlighted these.

3) In section 2, it can be found that many studies oriented to IoT security are based on elliptic curves, what’s the problem with the existing solutions? Besides, there should be numerous IoT security-related works based on blockchain, please discuss related typical prior arts

Answer: we added a discussion about the limitations of the related work in section 2 (also, these were already summarized in section 5.2). We added reference [36], study that also involves blockchain and elliptic curves for distributed vehicular systems.

4) In 4.1, for the high scalability of the authentication framework, the blockchain should be a distributed system. However, what is the relationship between each blockchain entity and IoT devices, and how does each IoT device find the right blockchain node? In Figure 1, the blockchain looks like a single node, but it is not true.

Answer: we added a description of how an IoT device should find the right blockchain node. Of course, the blockchain network has multiple nodes (we also modified Figure 2 – it was Figure 1 in the previous version of the manuscript – to illustrate this). We added section 4.2 which shows the architecture of the proposed system and explains the interaction between the components.

5) In 4.2, please highlight what’s the difference between the proposed algorithm and the ECC. In addition, it is better to make a complexity analysis of the proposed algorithm

Answer: We added the time complexity analysis at the end of section 4.2.

6) In section 6, please the detailed simulation environment setting in case the performance of different approaches is evaluated by simulations.

Answer: We added the information about simulation environment settings. Also, we detailed in section 6 the results that we obtained.

Round 2

Reviewer 2 Report

The authors addressed my previous concerns. The authors should check the writing before acceptance.

Author Response

Dear Reviewer,

Thank you for your positive feedback. We checked the writing again.

Reviewer 3 Report

Most of the previous comments have been well-addressed. The quality of the paper has been significantly improved. I have no other comments, suggest accepting the paper after minor revisions. 

Author Response

Dear Reviewer,

Thank you for your positive feedback. We checked the writing and did minor changes in the description of the algorithms.